# The Ups and Downs of STAT Inhibition in Acute Myeloid Leukemia

**DOI:** 10.3390/biomedicines9081051

**Published:** 2021-08-19

**Authors:** Bernhard Moser, Sophie Edtmayer, Agnieszka Witalisz-Siepracka, Dagmar Stoiber

**Affiliations:** Department of Pharmacology, Physiology and Microbiology, Division Pharmacology, Karl Landsteiner University of Health Sciences, 3500 Krems, Austria; bernhard.moser@kl.ac.at (B.M.); sophie.edtmayer@kl.ac.at (S.E.); agnieszka.witalisz@kl.ac.at (A.W.-S.)

**Keywords:** STAT, JAK, acute myeloid leukemia, AML, tyrosine kinase inhibitor, FLT3

## Abstract

Aberrant Janus kinase-signal transducer and activator of transcription (JAK-STAT) signaling is implicated in the pathogenesis of acute myeloid leukemia (AML), a highly heterogeneous hematopoietic malignancy. The management of AML is complex and despite impressive efforts into better understanding its underlying molecular mechanisms, survival rates in the elderly have not shown a substantial improvement over the past decades. This is particularly due to the heterogeneity of AML and the need for personalized approaches. Due to the crucial role of the deregulated JAK-STAT signaling in AML, selective targeting of the JAK-STAT pathway, particularly constitutively activated STAT3 and STAT5 and their associated upstream JAKs, is of great interest. This strategy has shown promising results in vitro and in vivo with several compounds having reached clinical trials. Here, we summarize recent FDA approvals and current potential clinically relevant inhibitors for AML patients targeting JAK and STAT proteins. This review underlines the need for detailed cytogenetic analysis and additional assessment of JAK-STAT pathway activation. It highlights the ongoing development of new JAK-STAT inhibitors with better disease specificity, which opens up new avenues for improved disease management.

## 1. Introduction

Acute myeloid leukemia (AML) is a heterogeneous malignant disorder characterized by abnormal proliferation and differentiation of hematopoietic stem and progenitor cells (HSPCs). This clonal expansion leads to an accumulation of immature myeloid precursors in the hematopoietic tissue and peripheral blood, which occurs at the expense of normal hematopoiesis and results in anemia, leukopenia or thrombocytopenia [1,2]. The global AML burden increases with the ageing population, because the incidence of AML increases with age and is predominantly diagnosed in older patients with a median age at diagnosis of 68 years [3]. The American Cancer Society estimated that AML will be responsible for 11,400 deaths in the United States in 2021, accounting for the highest percentage (48%) of leukemic deaths, with a rate of relative 5-year survival (2010–2016) of 68% (age < 20) and 8.2% in patients aged 65 years and over [4]. The development of intensive consolidation chemotherapy programs and progress in supportive care have ameliorated the survival rate of younger adults with AML (aged < 60 years); however, the survival rate in the elderly has failed to show a substantial improvement over the past decades [5,6]. The characterization of distinct AML subtypes by multiple cytogenetic and molecular abnormalities is used to better predict the prognosis of patients and to make treatment decisions under refined risk stratification. Recent advances in next-generation sequencing technology play a major role and provide an in-depth insight into the mutational driver landscape of AML, which will continue to become increasingly important in assessing the risk of AML progression. Identification of new unique biomarkers associated with acute leukemias significantly improves the prognosis and diagnosis of genetically unique AML patients and improves the diagnostic criteria [7,8,9,10]. The current World Health Organization (WHO) classification considers morphologic, cytogenetic and molecular genetic analysis, clinical history, and prognostic and immunophenotypic data. It defines specific disease entities by focusing on genetic subgroups: AML with recurrent genetic abnormalities (AML-RGA), AML with myelodysplasia-related changes (MDS-AML), therapy-related AML (tAML) and AML not otherwise specified (AML-NOS) [11]. Analysis of the Surveillance, Epidemiology, and End Results (SEER) database, containing 32,941 AML patients, showed that AML patients who were not diagnosed according to WHO classification lived shorter than patients diagnosed using WHO-type classification [12]. This clearly reflects that many AML patients may not receive an optimal treatment and underpins the benefits of targeted AML therapy. Each patient’s AML is genetically unique, since individual leukemias can result from distinct mutations with even more diverse subclones. More than 86% of AML patients harbor two or more driver mutations, while only 3% do not contain any known driver mutation [13]. Most frequent alterations appear in genes expressing signaling proteins (e.g., *KIT*, *FLT3*, *NPM1*, *CEBPA*), epigenetic modulators (e.g., *EXH2*, *TET2*, *DNMT3A*) or as chromosomal rearrangements that generate fusion transcripts of genes encoding for transcription factors. These include, for example, *RUNX1-RUNX1T1*, *PML-RARA* or *MLLT3-MLL* [14,15,16,17,18,19,20,21].

The Janus kinase (JAK) family of nonreceptor tyrosine kinases comprises pivotal mediators of growth factor and cytokine signaling, activating i.a. downstream signal transducer and activator of transcription (STAT) factors [22,23,24]. STATs, particularly STAT3 and STAT5, are hyperactivated in various solid tumors and hematological malignancies stimulating cellular proliferation and survival [25,26,27,28,29,30,31]. STAT3 and STAT5 are constitutively activated in 44–76% of AML patients [32,33,34] and individuals show a shorter disease-free survival with constitutive STAT3 activity than without [33]. Interestingly, our group demonstrated that the alternatively spliced STAT3β isoform has a tumor-suppressive role and that the *STAT3*β/α mRNA expression ratio positively correlates with overall survival, representing a promising prognostic marker in AML patients [35]. Despite significant advances, the exact contribution of JAKs and STATs to the development of AML remains to be discovered. Current trends in ongoing research are reflected in the approaching progress of drug development and their recent approvals. As a consequence, especially for STAT3 and STAT5, the upstream JAKs and co-factors have become attractive therapeutic targets in AML. Here, we present a comprehensive overview of clinically relevant inhibitors for AML patients, acting on JAK-STAT signaling.

## 2. JAK-STAT Signaling

### 2.1. The JAK-STAT Signaling Pathway

In mammals, four JAK proteins (JAK1, JAK2, JAK3 and tyrosine kinase 2 (TYK2)) and seven STAT proteins (STAT1, STAT2, STAT3, STAT4, STAT5A, STAT5B and STAT6) have been identified. The binding of a ligand to its receptor is a critical step in precisely determining the context-dependent effect of downstream signaling by distinct JAK-STAT combinations with a high degree of specificity. The JAK-STAT signaling pathway links the receptor binding of extracellular ligands like cytokines, hormones and growth factors via a ubiquitous intracellular signaling cascade with the transcriptional machinery in the nucleus. It is tightly regulated under physiological conditions coordinating a wide range of fundamental biological mechanisms, including cell proliferation, differentiation, apoptosis and inflammation. Dysregulation has been identified in severe pathological circumstances and is commonly associated with constitutively active STATs promoting, among others, intense inflammatory conditions or tumorigenesis [36].

The highly conserved JAKs, JAK1, JAK2 and TYK2, are ubiquitously expressed, whereas JAK3 is predominantly expressed in hematopoietic cells, in particular lymphoid and myeloid cells [37]. Among tyrosine kinases, JAKs are unique for their non-covalent interaction via N-terminal 4.1, ezrin, radixin, moesin (FERM) and Src Homology 2 (SH2) domains with class I and II cytokine receptors [38]. The C-terminal region contains the active tyrosine kinase and the pseudokinase domain, which regulates both basal and cytokine-induced activation [39,40]. Mechanistically, the JAK-STAT signaling pathway (Figure 1) is activated by ligand binding to their cognate multimeric transmembrane receptors (such as interleukin (IL), interferon (IFN) and hormone receptors) leading to conformational changes of the receptor. Receptor-associated JAKs are now in close proximity, inducing their activation by autophosphorylation and/or transphosphorylation and subsequently the phosphorylation of the receptor-associated tyrosine residues. This in turn provides a docking site for the SH2 domain of STAT proteins and leads to their JAK-mediated phosphorylation of a conserved C-terminal tyrosine residue. Subsequently the phosphorylated tyrosine of one receptor-associated STAT reciprocally binds the SH2 domain of another receptor-associated STAT [36,41]. This anti-parallel to parallel homo- or heterodimerization of phosphorylated STAT (pSTAT) proteins ensures the release from the receptor and the translocation into the nucleus by importins. STAT2, STAT3, STAT5, and STAT6 translocate constantly to the nucleus independent of tyrosine phosphorylation, which only transiently increases their nuclear accumulation. In the nucleus, STAT proteins control the transcriptional activation and repression of associated genes by directly targeting DNA regulatory elements (DRE), which are usually found in the first introns of their target genes and in enhancer and promoter regions. Through chromatin immunoprecipitation followed by sequencing (ChIP-seq) and the integration of gain- or loss-of-function transcriptomics, it is possible to identify various STAT binding sites and target genes; e.g., *MYC*, *BCL-2*, and *D-type cyclins*, which are engaged by any STAT protein in various cell types [36,42].

### 2.2. JAK-STAT Activation

Based on the ligand, the receptor and their tissue specificity, distinct combinations of STATs and JAKs are activated. This canonical signaling is based on JAK-mediated STAT tyrosine-phosphorylation. Additionally, several non-canonical pathways have been described, including cascades building on the kinase-independent action of JAKs and unphosphorylated STATs, or the formation of higher-order or multifactorial STAT transcriptional complexes, which can also affect cytokine signaling and immune cell function [36,43]. The most common non-canonical pathway activators include, among others, the cytoplasmic non-receptor tyrosine kinases SRC and ABL (or BCR–ABL fusion proteins in Philadelphia chromosome (Ph^+^) leukemias), the platelet-derived growth factor receptor (PDGFR) and epidermal growth factor receptor (EGFR) [41,44,45]. Compared to the genomic targets of the canonical STAT signaling pathway, which are predominantly composed of genes implicated in the development of cellular immunity, target sites of STATs of the non-canonical pathway were found to be rather associated with genes implicated in epigenetic regulation [46].

During canonical STAT signaling, STATs act mostly as homodimers, but they are also found as heterodimers or higher-order tetramers [36]. STAT1 and STAT2 are involved in IFN signaling. While STAT1 is implicated in anti-viral and -bacterial defense, tumor-growth inhibition and apoptosis, STAT2 participates in immunomodulatory and anti-apoptotic/-proliferative signaling. JAK1 together with JAK2 is induced by IFN-γ and JAK1 together with TYK2 by type I and type III IFNs, making JAK1 imperative for IFN signaling [43]. IFN-γ induces exclusively the phosphorylation of STAT1, leading to its homodimerization and finally binding to gamma interferon-activated sites (GAS) of IFN-regulated genes. Type I IFN target genes are induced by heterodimers of STAT1 and STAT2, which, together with IRF9, form the interferon stimulated gene factor 3 (ISGF3) complex that regulates transcription via binding to interferon-stimulated response elements (ISREs) [41,47,48,49]. STAT3 is activated by numerous cytokines, including IL-6, IL-10 or hormones like prolactin and leptin, as well as growth factors like EGF, PDGF, fibroblast growth factor (FGF) and insulin-like growth factors (IGFs) [50]. It is engaged in immunity and inflammation but also in multiple other biological processes including immune response resolution, wound healing [51] and restoration of tissue integrity [52]. For example, IL-6 binds the IL-6 receptor and glycoprotein 130 (gp130) receptor β-subunit complex and activates JAK1, JAK2 and TYK2, leading to the formation of STAT3 homodimers [53,54,55]. Aberrant STAT3 signaling drives mitogenesis and anti-apoptotic pathways and furthermore tumorigenesis and metastasis [50]. STAT4 is inevitable for cellular-mediated immune response by regulating differentiation of Th1 cells, lymphocyte effector function and IFN-γ production [56,57]. IL-23 and IL-12 signal through JAK2 and TYK2 while IL-23 induces STAT3/STAT4 heterodimers and IL-12 signaling results in STAT4 homodimers [58]. Signaling via the two highly related STAT5 proteins, STAT5A and STAT5B, occurs primarily through JAK1 and JAK3 upon binding of IL-15, IL-7 and IL-2 to their respective receptors and the γc family of receptors [59]. Furthermore, STAT5 can be activated in a JAK2-dependent manner by prolactin (PRL), growth hormone (GH), erythropoietin (EPO) and granulocyte-macrophage colony-stimulating factor (GM-CSF) [60]. STAT5A is predominantly expressed in the mammary gland where it is involved in PRL signaling, whereas STAT5B is mainly expressed in the liver and muscle, regulating GH signaling [61]. STAT5 is a key transcription factor contributing to the initiation and progression of various cancers, but under physiological conditions STAT5 signaling is imperative for hematopoiesis [62], lipid metabolism [63] and T lymphocyte [64] and natural killer (NK) cell development and function [65]. Similar to STAT5, STAT6 is activated by JAK1 and JAK3, but activation occurs via binding of IL-4 and IL-13 to their receptors [66]. It is involved in the development of allergic inflammation, Th2-type responses [67], B and T cell proliferation and macrophage differentiation [68,69]. Besides classical STAT6 homodimerization, type I IFN-stimulated B cells induce the heterodimerization of STAT6 and STAT2 [70], whereas IL-4 promotes STAT6/STAT2 heterodimers in T cells [71].

### 2.3. JAK-STAT Inactivation

The JAK–STAT signaling cascade is subject to constant multiple levels of activating and inactivating processes, which offer new possibilities for increased therapeutic exploitation. Pathway inactivation can be accomplished by basic cellular mechanisms including posttranslational modifications, microRNAs, RNA-binding proteins [36], the ubiquitin-proteasome system [72] or receptor internalization through endocytosis [73]. Various protein tyrosine phosphatases (PTP) including Src (Sarcoma) Homology 2 Domain Phosphatase 1 and 2 (SHP1, SHP2), CD45 and T-cell protein tyrosine phosphatase (TCPTP), just to name a few, can hydrolyze phosphorylated JAKs, upstream receptors and STATs, both in cytoplasm as well as in the nucleus [74]. Protein inhibitor of activated STAT (PIAS) proteins are activation-suppressing proteins, which negatively regulate JAK-STAT signaling by inhibiting STAT phosphorylation, translocation or DNA binding [75]. The protein family of suppressors of cytokine signaling (SOCS) acts via negative feedback circuits. All eight SOCS proteins (SOCS1-7 and CIS) contain SH2 domains to interact competitively with receptors [76,77]. Additionally, SOCS can recruit E3 ubiquitin ligases to induce Lysin 48-linked polyubiquitylation and subsequent proteasomal degradation of the receptor or the JAKs [78]. SOCS can also target JAKs’ substrate binding groove with high specificity, via their N-terminal kinase inhibitory region, which functions as a pseudo-substrate for JAKs [79].

### 2.4. Dysregulation of JAK-STAT Signaling in the Pathogenesis of AML

The complex network between genetic factors and extrinsically driven signaling pathways is crucial for tumorigenesis and other severe diseases. JAK-STAT signaling in the hematopoietic system tightly orchestrates survival and self-renewal of hematopoietic stem cells (HSC) and hematopoiesis, as well as proliferation. Thus, aberrant JAK-STAT signaling has been associated with the pathogenesis of inflammatory diseases and immunodeficiencies and also with hematopoietic malignancies [80,81]. Since the 1990s, mutations of proteins inducing the constitutive activation of JAK-STAT have been associated with AML [30,82,83,84]. Importantly, constitutive activation of STAT3 and STAT5, and also of STAT6, can function as a predictive and prognostic biomarker and has been correlated with severe disease outcomes in AML [26,84].

Constitutive activation of STATs has often been found in cancer cells as a result of deregulated signaling of the upstream JAK. A classic example is *JAK2*^V617F^, a somatic point mutation within the autoinhibitory pseudokinase (JH2) domain, resulting in constitutive activation of the JH1 domain and thus increased activation of JAK2 and downstream STATs [85]. *JAK2*^V617F^ appears in the majority of secondary AML cases resulting from myeloproliferative neoplasm (MPN), where it acts as a main driver mutation. Mutated JAK2 is only present in <5% of de novo AML patients [23,24].

In de novo AML, constitutive activation of STATs can be achieved by auto- and paracrine factors such as IL-6, decreased expression of negative regulators such as SOCS1 or activating mutations in other (non-JAK) upstream kinases such as FMS-like tyrosine kinase 3 (*FLT3*) [86]. *FLT3* mutations, especially internal tandem duplication mutations (ITDs) of the extracellular juxtamembrane (JM) domain, are the most frequent genetic alterations in AML patients with a normal karyotype [13,87]. FLT3 regulates proliferation, differentiation and survival in both myeloid and lymphoid lineage development through STAT5 activation, as well as via PI3K and RAS signaling [88]. The FLT3-ITD induces constitutive activation of STAT5 independently of upstream JAK2 or SRC and is resistant to SOCS inhibition. While FLT3-WT activates STAT5 indirectly via the ERK pathway, the mutated FLT3 directly phosphorylates STAT5 [89].

*FLT3* mutations frequently co-occur with alterations in the *NPM1* gene, leading to a significantly worse prognosis [90]. Mutated NPM1 can translocate mutated FLT3 to the endoplasmic reticulum and enhance STAT5 activation [91], as well as *MYC* and *BCL2* expression in a bromodomain-containing protein 4 (BRD4)-dependent manner [92]. However, phosphorylated STAT5 downregulates NPM1 expression via BRCA1-BARD1 ubiquitin ligase signaling in human erythroleukemic cells. In addition to the crucial role of STAT5, STAT3 also interacts with NPM1 and can transcriptionally enhance its expression [90]. *FLT3*-ITDs also combine with mutations of *DNMT3A*, which seems to lead to the worst prognosis for AML patients [13,93]. DNMT3A preserves DNA methylation patterns during replication [94] and mutated DNMT3A elevates the self-renewal capacity of HSC [95]. Increased DNMT3 activity has been correlated with activation of STAT5 and hypermethylation of the *PTEN* promoter in AML cells [96].

Another way in which the JAK-STAT pathway contributes to the pathogenesis of AML is the reliance of leukemic cells on its signaling. AML cells carrying *CEBPA* or *CEBPA/CSF3R* mutations are sensitive to the inhibition of the JAK-STAT pathway [97]. Similar sensitivity has been detected in AML cells expressing *RUNX1–RUNX1T1* [98]. Furthermore, the gain of function mutation of the tyrosine kinase c-KIT (*KITD*^D816V^) is frequently detected in AML patients [99,100] and is strongly associated with STAT3/STAT5 activation [26]. These data and several other studies highlight the role of STAT transcription factors, especially STAT3, STAT5A and STAT5B, in AML. Our STRING analysis illustrates that among all STATs, STAT3 and STAT5 show the strongest connections to the most frequent AML drivers, underlining their contribution to AML pathogenesis (Figure 2). We therefore describe current approaches directly targeting STAT3 and STAT5 and their upstream kinases (JAKs, FLT3) in pre-clinical and clinical studies for AML.

## 3. JAK Inhibitors

Inhibition of JAK-STAT signaling via JAK1/2 inhibitor Ruxolitinib is a successful approach in the treatment of MPNs. In rare cases MPNs evolve to secondary AML (sAML), which provided the first rationale for investigating Ruxolitinib as a treatment for sAML [103,104,105,106,107]. The understanding of the mutational landscape and its signaling consequences led to the identification of hyperactivity of the JAK-STAT pathway in many AML subtypes, independent of the *JAK2*^V617F^ mutation [86]. Therefore, Ruxolitinib and other JAK inhibitors are nowadays tested in AML preclinical settings with some having reached clinical trials (Table 1, Figure 3).

Ruxolitinib has been approved for the treatment of intermediate and high-risk MPNs that are resistant or intolerant to hydroxyurea. In MPN, inhibition of JAK1/2 as a monotherapy efficiently reduces symptoms [107]. Unfortunately, in the heterogeneous AML patient cohort, the anti-leukemic activity of Ruxolitinib as a single agent has been rather poor [121,122]. However, preclinical models of pediatric acute megakaryoblastic leukemia (AMKL) with enhanced activity but no known driver mutation in the JAK-STAT pathway revealed Ruxolitinib as an effective drug [123]. This suggests that Ruxolitinib’s treatment efficacy as single agent may be improved by a more personalized approach. Another strategy to increase the therapeutic effect of Ruxolitinib in AML is the combination with other agents. Decitabine, a DNA methylation inhibitor [124], was discovered to be very effective in combination with Ruxolitinib in a mouse model of AML, driven by genetic knockout of p53 and *JAK2*^V617F^ mutation [125]. This combinatorial approach was further implemented in clinical trials where it demonstrated a promising 29–42% of complete remission (CR) and complete remission with incomplete hematologic recovery (CRi) during phase I/II trials in patients with post-MPN AML. However, the overall survival remained poor [126,127,128]. Final results of the clinical trial are still awaited; however, the authors suggest that this approach could be an alternative to aggressive chemotherapy for obtaining CR prior to allogenic hematopoietic stem cell transplant [127]. A recent approval of a combination of Decitabine with BCL2 inhibitor Venetoclax, which is also directed to AML patients who are not eligible for intensive chemotherapy, diminishes the attractivity of this novel combinatorial approach of Ruxolitinib and Decitabine [127,129]. In particular, preclinical studies have shown promising results upon combining Venetoclax with Ruxolitinib. In patient samples and xenograft mouse models Ruxolitinib can reverse the resistance to Venetoclax induced by bone marrow (BM)-stromal cells [130].

Among many combinatorial approaches being tested in preclinical settings, a combination of Ruxolitinib with Lysine Specific Demethylase 1 (LSD1) inhibitor recently demonstrated a synergistic survival improvement in mouse models of *CEBPA*- and *CSF3R*-mutated AML [97]. LSD1 demethylates histones H3K4 and H3K9, resulting in transcriptional regulation, and its inhibition alone induces differentiation in a subset of AML cell lines [131]. The effect is profoundly enhanced by combination with Ruxolitinib [97].

Lestaurtinib is a multikinase inhibitor initially developed as a tropomyosin receptor kinase A (TrkA) inhibitor, which also potently inhibits JAK2 and FLT3 [132]. *FLT3*-mutated AML cells are sensitive to Lestaurtinib in vitro upon treatment with chemotherapeutic agents [133,134]. The same approach was shown to be effective in vivo in a murine model of *FLT3*-ITD AML [134]. On this basis, Lestaurtinib was assessed in clinical trials in combination with classical chemotherapy in patients with relapsed *FLT3*-mutated AML. Although a transient response appeared in treated patients, overall, no clinical efficacy was observed in this therapeutic setting [135].

Pacritinib is a multikinase inhibitor with high specificity towards JAK2 over other JAKs, but in addition it inhibits interleukin receptor-associated kinase (IRAK), Colony stimulating factor 1 receptor (CSF1R) and FLT3 [136]. It is currently undergoing phase II/III clinical trials for MPNs. Pacritinib not only effectively inhibits the growth of many *FLT3*-ITD+ AML cell lines, but combined inhibition of JAK2 and FLT3 also overcomes the resistance towards FLT3 inhibitors (e.g., Sunitinib), which is often associated with overactivation of the JAK-STAT pathway [137]. Interestingly, a more recent report identified Pacritinib as being effective in AML mouse models and patient samples irrespective of the mutational status of *FLT3* and *JAK2*. The therapeutic effect was attributed to efficient IRAK1 inhibition rendering this kinase a novel target in AML [138]. Indeed, a combined treatment regimen of Pacritinib and chemotherapy was well tolerated and demonstrated preliminary anti-leukemic activity in patients with *FLT3* mutations [139].

Fedratinib is another small molecule targeting JAK2 and FLT3, which has recently been approved for the treatment of MPN. In addition, Fedratinib inhibits BRD4 [140]. Preclinical studies identified Fedratinib to be effective in reducing the leukemic burden in xenograft models as a single agent and resulting in a stronger anti-leukemia response in combination with cytarabine [141]. Similarly to Ruxolitinib, Fedratinib is currently undergoing phase II clinical trials in AML in combination with Decitabine, but no results are available yet.

Momelotinib is a JAK1/2 inhibitor that has reached phase III clinical trials for MPNs [142,143]. Although limited data exist concerning the effectiveness of Momelotinib in AML, it has been shown to effectively inhibit the growth of AML cell lines and disease progression in a xenograft model. Interestingly, the anti-leukemic effect was attributed to the inhibition of the kinase IKBKE, a noncanonical IkB kinase, which drives MYC expression [144]. Momelotinib, similarly to other JAK2 inhibitors, also shows inhibitory activity against FLT3. A recent study using AML cell lines and mouse models of *FLT3* mutated AML suggests that Momelotinib-mediated combinatorial inhibition of JAK1/2, FLT3 and downregulation of MYC will be more effective than currently used FLT3 tyrosine kinase inhibitors (TKIs) [145].

Inhibition of JAKs in combination with chemotherapy or epigenetic modulator inhibitors seems to be a promising approach in the treatment of different subtypes of AML. In contrast to MPNs, where the specific JAK2 inhibitors sparing other kinases are discussed as the best treatment option [146], AML patients seem to benefit more from multikinase inhibitors, which, in addition to JAK-STAT, target another pathway(s). This approach is especially successful in *FLT3* mutated AML. Although occurring with significantly lower incidences, many other mutations in AML drive constitutive activation of the JAK-STAT pathway [86]. Therefore, further studies identifying new targets resulting in synergistic effects and finding new combinatorial inhibitors are required.

## 4. STAT Inhibitors

A substantial proportion of AML patients harbor mutations that cause hyperactivation of JAK-STAT signaling [147]; therefore, great effort has been put into the development of inhibitors that block phosphorylation mediated by JAKs, which have already been shown to efficiently achieve indirect STAT inhibition [86]. However, many TKIs showed off-targets besides JAKs, potentially causing more side effects that could be prevented by targeting downstream proteins relevant for leukemogenesis [148]. Furthermore, JAK-independent mechanisms activating STATs were also identified [149,150,151]. Therefore, direct interference with STATs remains an attractive therapeutic approach. Unlike other STAT family members, STAT3 and STAT5 have been extensively studied in hematopoietic malignancies and were shown to be key factors in AML. In most AML patient samples of peripheral blood or bone marrow, constitutive activation of STAT3 and/or STAT5 was observed [30,32,33,152]. Therefore, we will focus on those STAT family members and summarize the current strategies in the development of inhibitors in the context of AML (Table 2, Figure 2 and Figure 3).

### 4.1. STAT3

STAT3 has been shown to play an important role during myeloid homeostasis and differentiation as well as proliferation [161,162]. Due to its ability to block myeloid differentiation, its crucial role during leukemogenesis is highlighted [154,163]. Target genes of STAT3 driving survival and proliferation include *MYC*, *cyclin D1*, *BIRC5* (Survivin) and *BCL2.* In line, upregulation of STAT3 is associated with protection from apoptosis [164]. Furthermore, constitutive STAT3 activity is associated with adverse patient outcomes [165] and significantly reduced disease-free survival in AML patients [33]. STAT3 signaling is frequently altered due to activating mutations [166,167] or constitutive stimulation of upstream signaling [34]. Based on those findings, STAT3 has become an attractive therapeutic target in AML.

A small molecule targeting the SH2 domain of STAT3, OPB-51602, has entered clinical trials for the treatment of hematological malignancies. It was shown to efficiently repress proliferation by blocking the phosphorylation sites tyrosine705 and serine727 of STAT3 in various human cancer cell lines [168]. More recent studies revealed that the inhibitor in particular interferes with the mitochondrial function of STAT3, which was demonstrated in human cancer cell lines in vitro [169]. In this phase I study, 20 patients with relapsed or refractory hematological malignancies were treated daily based on the “3 + 3” design. Besides being safe and well tolerated, common adverse events such as nausea, peripheral sensory neuropathy and diarrhea were reported. Nevertheless, no clear therapeutic response was obtained in this patient cohort, except two AML patients and one myeloma patient that were reported with durable stable disease. However, the authors claim that the optimal dose and daily dosing schedule for long-term administration was difficult to determine in this heterogeneous patient cohort, which thus suggests a need for further investigation in a different patient group [170].

The small molecule STAT3 inhibitor, C188-9, was demonstrated to inhibit G-CSF-induced STAT3 activation thereby efficiently inducing apoptosis in primary pediatric AML samples and AML cell lines [154]. The compound targets the phosphotyrosine (pY) peptide binding site within the SH2 domain, subsequently preventing the interaction with tyrosine kinases and dimerization. However, further studies mainly focused on its antitumor activity in solid cancer such as head and neck squamous cell carcinoma [171] and pancreatic cancer [172] demonstrating its potential to impair tumor growth in vitro.

BBI608, also known as Napabucasin, is a small molecule inhibitor blocking STAT3-mediated gene transcription, whose antitumorigenic properties have been demonstrated in phase Ib/II and phase II trials in solid cancers as a monotherapy and in combination with standard treatment [173,174]. Recently, BBI608 was also shown to exhibit anti-leukemic effects in human AML cell lines in vitro as well as in primary AML samples obtained from 21 patients. In this study, BBI608 treatment led to a promising decrease in tumor burden in an in vivo AML xenograft model using the human AML cell line MOLM-13. Furthermore, combination treatment of BBI608 together with Venetoclax demonstrated enhanced cytotoxicity in BBI608-resistant Kasumi-1 cells [175]. Taken together, these data suggest BBI608 as a potential candidate for treatment in AML; however, this requires further evaluation.

An emerging strategy to target proteins with small-molecule degraders are proteolysis targeting chimeras (PROTACs), which rely on hijacking the cellular degradation machinery, the proteasome. By using a heterobifunctional degrader, the protein of interest is linked to an E3 ligase, which subsequently leads to polyubiquitination and degradation of the target [176]. Due to this interplay, PROTACs are considered to have high selectivity. Recently, a PROTAC called SD-36 was shown to selectively induce degradation of STAT3 over all other STATs in vitro and in vivo [155]. SD-36 consists of lenalidomide, a CRBN ligand analog, and the STAT3 inhibitor SI-109, which recognizes the SH2 domain of STAT3. In this study, SD-36 efficiently degrades STAT3 in vitro in various leukemia and lymphoma cell lines. Besides inducing tumor regression in a MOLM-16 dependent xenograft tumor model, it was also well tolerated in the in vivo setting. SD-36 is the first PROTAC targeting STAT3, which demonstrates that modestly selective inhibitors such as SI-109 can also be improved by using advanced technologies.

An attractive strategy that emerged with the fast progress of next-generation sequencing techniques involves antisense oligonucleotide (ASO) inhibitors, which are designed based on the genetic sequence of the target gene. One promising STAT3 ASO, AZD9150, showed efficiency when systemically administered in preclinical trials for lymphoma and lung cancer (NSCLC) patients [156]. In vitro, AZD9150 achieved a decrease in STAT3 expression without affecting STAT1 and STAT5 levels in various human cancer cell lines. A further phase Ib trial with 30 relapsed or refractory non-Hodgkin’s lymphoma patients was undertaken, which identified DLBL patients as the best responding group [157]. Recently, AZD9150 was shown to promote hematopoietic differentiation in primary MDS and AML samples, also supporting its rationale for AML [177].

As a driver of immune evasion in cancer it is conceivable that inhibition of STAT3 enhances anti-tumor immune responses. Thus, Hossain et al. investigated small interfering RNAs (siRNAs) conjugated to cytosine guanine dinucleotides (CpG) that target *STAT3* gene silencing. To deliver the inhibitor to Toll-like Receptor (TLR)-9 positive antigen-presenting immune cells, the inhibitor molecule was linked to a TLR9 ligand and a CpG [158]. Administration of CpG-STAT3 siRNA in a mouse model mimicking human inv(16) AML resulted in disease regression in a CD8+ T-cell-dependent manner. Furthermore, STAT3 silencing/TLR9 triggering led to enhanced immunogenicity of primary AML cells. Based on this immunostimulatory strategy, the decoy oligodeoxynucleotide (dODN) inhibitor CpG-STAT3dODN was shown to block transcriptional activity of STAT3 by acting as a DNA decoy molecule sequestering STAT3 in the cytoplasm. Xenografts using 14 AML patient-derived samples showed efficient immune-mediated eradication by CD8/CD4+ T cells in mice. However, it could not be excluded that CpG-STAT3dODN can also partially block STAT1 [159]. Nevertheless, this strategy based on the enhancement of T cell response might not only be of clinical relevance for AML but also for other hematological malignancies.

### 4.2. STAT5

STAT5 plays a pivotal role especially in AML patients who harbor fusion-oncogenes such as *BCR-ABL* and *FLT3*-ITD, which contribute to its constitutive activation [32]. For AML patients harboring *FLT3*-ITD, which is one of the most commonly detected cytogenetic abnormalities, several upstream inhibitors targeting FLT3 kinase achieved promising clinical outcomes [178]. Thus, the majority of pharmacological interventions targeting STATs have focused on STAT3 and not STAT5. Two of those TKIs, namely Midostaurin and Gilteritinib, were also approved by the FDA for the treatment of *FLT3* mutated AML [179,180]. Although FLT3 inhibitors are nowadays part of standard therapy in *FLT3* mutated AML patients, the occurrence of therapy resistance is an ongoing challenge. Thus, using several approaches to target a disrupted signaling pathway might be an efficient strategy to eradicate leukemic cells and prevent therapy resistance.

In 2012, Page et al. performed an in vitro screen focusing on salicylic- acid-containing inhibitors potentially serving as phosphotyrosine mimicry to the SH2 domain of STATs [160]. Two of the lead compounds, BP-107 and BP-108, showed high binding affinity to STAT5 and efficiently suppressed its phosphorylation. Evaluation on the *BCR-ABL* positive K562 (CML) and *FLT3*-ITD positive MV-4-11 (AML) human cell lines revealed effective suppression of STAT5 phosphorylation and downregulation of target genes such as *Cyclin D1*, *Cyclin D2*, *C-MYC* and *MCL-1*. Further compounds, SF-1-087 and SF-1-088, were identified, which also showed high selectivity for STAT5 over STAT3 and STAT1.

As a novel SH2 inhibitor of STAT5, AC-4-130 was demonstrated to efficiently diminish STAT5 activity in human AML cells lines and primary *FLT3*-ITD-driven AML cells. In this preclinical setting, AC-4-130 exhibited anti-leukemic activity in vitro and in vivo in a xenograft model based on MV-4-11 cells by selectively targeting STAT5 over STAT1 and STAT3. Furthermore, it showed enhanced efficiency in combination with Ruxolitinib [27].

## 5. Conclusions and Outlook

To summarize, various strategies for the inhibition of deregulated JAK-STAT signaling have demonstrated promising results in vitro and in mouse models with some compounds reaching clinical trials for AML. However, the heterogenicity behind AML development and progression might be an explanation for the diverse treatment outcomes. To address this, a better understanding of the complex biology and the underlying mechanisms involved in JAKs and STATs modulating AML progression is needed. For example, a recent preclinical study identified strong dependency of rarely occurring MLL-AF10-driven AML on JAK1 [86]. Thus, personalized approaches and identification of predictive biomarkers might improve the therapeutic benefit from JAK inhibition. Due to their multifunctionality as transcription factors, STAT proteins also have divergent functions depending not only on the AML subtype. In the case of STAT3, it was shown that the two alternatively spliced isoforms can have opposite roles in leukemogenesis [35], making it even more challenging to predict the effects of STAT3 targeting. Further, as emerging chemoresistance in leukemic cells is an ongoing challenge, which is reflected in the high relapse rate, it is important to interfere with cancer drivers on different levels. As illustrated by the example of JAK inhibitors, the most successful compounds are those that in addition to JAKs target another pathway (e.g., FLT3). Such kinase inhibitors, as well as combinational treatment of JAK or STAT inhibitors with other compounds, would be the best strategy to eliminate leukemic cells efficiently before they develop resistance. Ruxolitinib-based combinatorial therapies are continuously increasing according to clinical needs. This is the best-studied JAK1/2 inhibitor, but it did not fulfill expectations as a single agent due to modest antileukemic activity in patients [121,122]. However, in combination with Decitabine [109,112,113,124] and also with Cytarabine [111], Venetoclax [114] or the STAT5 inhibitor AC-4-130 [27], it showed promising results and increased AML treatment efficacy. Overall, clinical trial efforts and financial incentives in the last decade have intensified the research on potential therapies and have led to a surge in potential clinical applications and approvals. Ongoing development of new JAK-STAT inhibitors with higher compound selectivity and better disease-specificity is central to the management of AML.

## Figures and Tables

**Figure 1 biomedicines-09-01051-f001:**
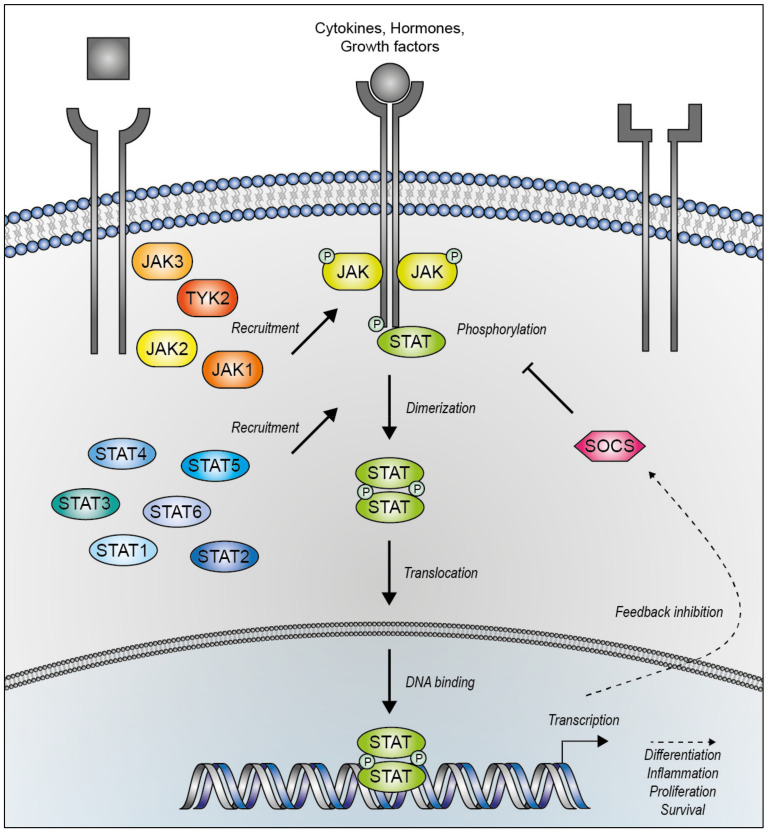
The JAK-STAT signaling pathway. The JAK-STAT pathway transmits signals from extracellular ligands (e.g., cytokines: IL-2, IL-3, IL-4, IL-5, IL-6, IL-7, IL-9, IL-10, IL-11, IL-12, IL-13, IL-15, IL-20, IL-21, IL-22, IL-23, IL-24, IL27, IL-31, IFNα, IFNβ, IFNγ, IFNλ, GM-CSF, G-CSF, leptin; hormones: EPO, PRL, GH, TPO; growth factors: EGF, insulin, FGF, PDGF, VEGF) through an intracellular signaling cascade to the transcriptional machinery in the nucleus. Ligand-binding leads to a conformational change of the receptor and the activation of receptor-associated JAKs by autophosphorylation and/or transphosphorylation and subsequently the phosphorylation of the receptor tyrosine residues. This in turn provides a docking site for the STAT proteins and leads to their JAK-mediated phosphorylation. Activated STATs translocate as either homo- or heterodimers to the nucleus and regulate gene transcription. Then, expressed SOCS proteins act via negative feedback circuits by interacting competitively with receptors. JAK (Janus kinase), P (phosphorylation), STAT (signal transducer and activator of transcription, SOCS (suppressors of cytokine signaling), TYK2 (tyrosine kinase 2).

**Figure 2 biomedicines-09-01051-f002:**
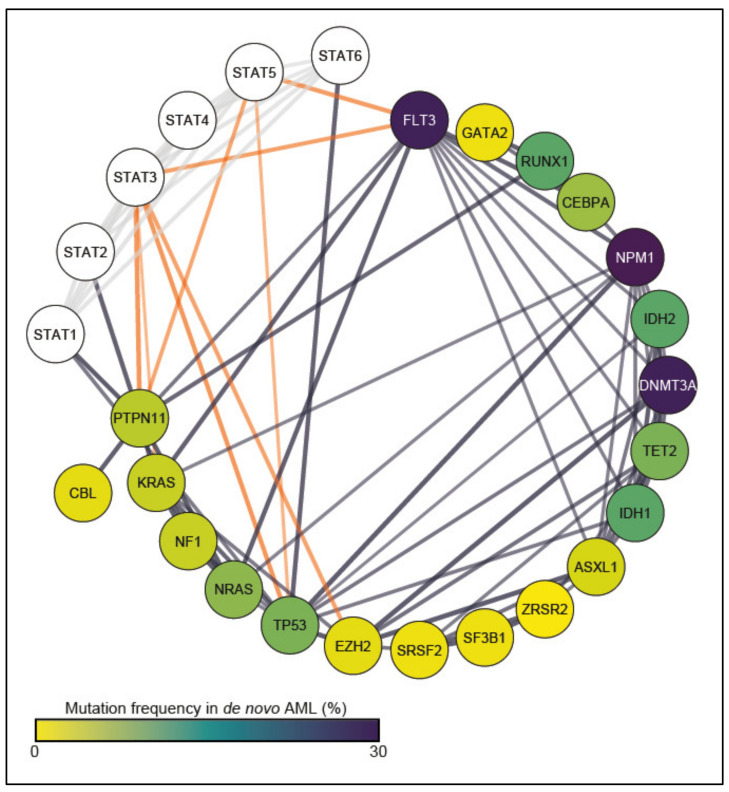
Protein–protein association analysis of STAT proteins and driver mutations in de novo AML. Visualization of STRING protein–protein interaction network of STATs and frequently mutated genes in de novo AML. Analysis includes physical as well as functional interactions of publicly available data [101]. Orange lines show connections of STAT3 or STAT5. The fill color of the nodes represents the mutation frequency of the genes [102]. The line width refers to the STRING interaction score with a confidence cut-off score of 0.7. Connections/protein–protein associations between STATs are illustrated in grey. STAT5A and STAT5B showed the same interactions and were therefore merged to STAT5 for clarity of the figure.

**Figure 3 biomedicines-09-01051-f003:**
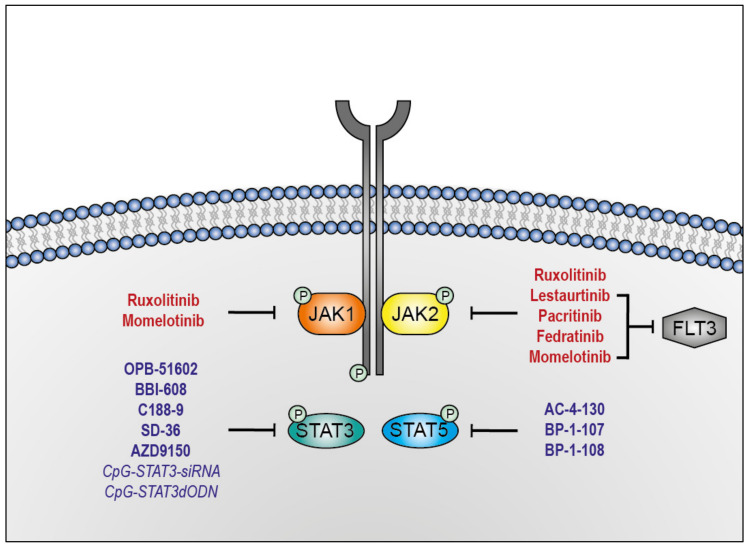
Targeting the JAK-STAT signaling pathway in AML. Representation of multiple JAK (red) and STAT (blue) inhibitors in clinical trials. Small interfering RNAs (siRNAs) and decoy oligodeoxynucleotide (dODN) are presented in italics.

**Table 1 biomedicines-09-01051-t001:** Current clinical trials of JAK inhibitors in AML.

	Drug	Diseases	Trials		References
**JAK1/JAK2**	Ruxolitinib (INCB18424)	AML, ALL, MDS, CML	Phase II		[108]
		MPN, AML	Phase II	Combination with Decitabine	[109]
		AML, ALL, MDS, CML	Phase II		[108]
		AML in CR	Phase II		[110]
		post MDS-AML	Phase I/II	Combination with Cytarabine	[111]
		relapsed/refractory post-MPN AML	Phase I/II	Combination with Decitabine	[112]
		MPN, post-MPN AML	Phase I/II	Combination with Decitabine	[113]
		post-MDS AML, recurrent/refractory AML	Phase I	Combination with Venetoclax	[114]
		Leukemia, MPD, solid tumor	Phase I		[115]
**JAK2/FLT3**	Fedratinib (TG101348)	MPN, AML	Phase II	Combination with Decitabine	[109]
	Lestaurtinib (CEP-701)	AML with FLT3 mutation	Phase II	Combination with standard treatment	[116]
		AML with FLT3 mutation	Phase II		[117]
		AML with FLT3 mutation	Phase I/II	Combination with standard treatment	[118]
	Pacritinib (SB1518)	AML with FLT3 mutation	Phase I	Combination with standard treatment	[119]
		AML, CML, MDS	Phase I/II	Monotherapy	[120]

AML (acute myeloid leukemia), CML (chronic myeloid leukemia), CR (complete remission), MDS (myelodysplastic syndrome), MPD (myeloproliferative disorder), MPN (myeloproliferative neoplasm). Clinical trials were searched on clinicaltrials.gov (June 2021) and included based on the following criteria: JAK inhibitors in AML, exclusion criteria: terminated, withdrawn, suspended, or status unknown.

**Table 2 biomedicines-09-01051-t002:** STAT inhibitors investigated in AML.

	Drug	Type	Target	Diseases	Trials		References
**STAT3**	OPB-51602	Small molecule	SH2 domain	AML, MM, NHL, ALL, CML	Phase I	Combination with Decitabine or Venetoclax	[120]
	BBI-608	Small molecule		MM, Lymphoma, AML, MDS, CML, CLL	Phase I	Monotherapy or in combination with standard therapy	[153]
	C188-9	Small molecule	SH2 domain	AML	Preclinical	Cell lines, primary cells	[154]
	SD-36	PROTAC	SH2 domain	AML, ALCL	Preclinical	Cell lines, xenografts	[155]
	AZD9150	ASO	mRNA	AML, MDS, DLBCL, HL, NHL	Preclinical	Cell lines, xenografts	[156,157]
	CpG-STAT3-siRNA	siRNA	mRNA	AML, MM	Preclinical	Cell lines, primary cells, mouse model, xenografts	[158]
	CpG-STAT3dODN	Decoy oligo-nucleotide	DBD	AML	Preclinical	Cell lines, mouse model, xenografts	[159]
**STAT5**	AC-4-130	Small molecule	SH2 domain	AML, CML	Preclinical	Cell lines, primary cells, xenografts	[27]
	BP-1-107 BP-1-108	Small molecule	SH2 domain	AML, CML	Preclinical	Cell lines	[160]

AML (acute myeloid leukemia), ALCL (anaplastic large cell lymphoma), ASO (antisense oligonucleotide), CML (chronic myeloid leukemia), CLL (chronic lymphocytic leukemia), DLBCL (diffuse large B cell lymphoma), HL (Hodgkin’s lymphoma), MDS (myelodysplastic syndrome), MM (multiple myeloma), NHL (non-Hodgkin’s lymphoma). Clinical trials were searched on clinicaltrials.gov (June 2021) and included based on the following criteria: STAT inhibitors in AML, exclusion criteria: terminated, withdrawn, suspended or status unknown.

## Data Availability

Not applicable.

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
