# Peer review of "The Ups and Downs of STAT Inhibition in Acute Myeloid Leukemia"

_biomedicines, 2021, doi:10.3390/biomedicines9081051_

Round 1

Reviewer 1 Report

The authors have described about the role of   STAT Inhibition in Acute Myeloid Leu- 2.  The review is well written and presents interesting and relevant information useful to modulate AML progression.

Author Response

We thank the reviewer for viewing our manuscript as well written and presenting interesting and relevant information.

Reviewer 2 Report

Line 19 page 2: Is "tumor protein 52 (TP53)" a typo? Please correct any other typos that might have appeared throughout the manuscript.
I do not find it necessary to name each and every abbreviation for the gene names.
Please use italics when depicting a gene name.
Please use italics when using the word "via".
Figure 1: Overall I like this figure, but please extend the arrows and the inhibition depictions. Also, please make sure that you have nice local centring where it is the case.
Line 16 page 7: Please use a space between the gene and the mutation.
Figure 2: Why is there only STAT5 and not STAT5A and STAT5B? If only STAT5 is available in STRING please mention this in the text.

Author Response

Line 19 page 2: Is "tumor protein 52 (TP53)" a typo? Please correct any other typos that might have appeared throughout the manuscript.

We apologize for this and any other typos appearing throughout the manuscript. We have corrected them.

I do not find it necessary to name each and every abbreviation for the gene names -

We now did not specify abbreviations which are mentioned as examples.

Please use italics when depicting a gene name. 
Please use italics when using the word "via".

We have corrected these mistakes throughout the manuscript.

Figure 1: Overall I like this figure, but please extend the arrows and the inhibition depictions. Also, please make sure that you have nice local centring where it is the case.

We adapted the figure accordingly.

Line 16 page 7: Please use a space between the gene and the mutation.

We thank the reviewer for pointing this out, we have corrected this mistake.

Figure 2: Why is there only STAT5 and not STAT5A and STAT5B? If only STAT5 is available in STRING please mention this in the text.

We thank the reviewer for this comment and added more information to the figure legend. STAT5A and STAT5B showed the same STRING scores and interactions in our analysis. Therefore, we merged it to STAT5 for clarity of the figure.               

Reviewer 3 Report

Aberrant janus kinase - signal transducer and activator of transcription (JAK-STAT) signaling is implicated in the pathogenesis of acute myeloid leukemia (AML). This review underlines the need for detailed cytogenetic analysis and additional assessment of JAK-STAT pathway activation, and furthermore it highlights the ongoing development of new JAK-STAT inhibitors.

This review is comprehensive and instructive for hematologists. There are no futher comments.

Author Response

We thank the reviewer for her/his effort/time and for claiming that our review is comprehensive and instructive for hematologists.

Reviewer 4 Report

The review by Moser and colleagues focusses on JA-STAT signaling in myeloid malignancies. The authors introduce into the topic and nicely summarize the currently available therapy options. However, there is a number of concerns in particular in regard to the more general part on JAK-STAT in hematopoiesis and myeloid malignancy.

  • The title should be reconsidered. One main area of JAK-STAT abnormalities are MPNs, which is also reflected in the manuscript.
  • Additional proof-reading by a native speaker recommended. Sentence structure sometimes quiet complicated
  • Introduction needs more structure. The overall intro of AML is relatively long and would benefit from some shortening
  • Most adult AMLs are CN, not fusion; signaling activation is found in all subgroups
  • Two hit model (Type I/II class) is outdated, even three classes may not do the job. Too much detail for a JAK/STAT focused review with too little connection to the topic
  • Stability of signaling mutations (de novo AML vs. relapse)?
  • Constitutive activation of STATs in AML (not by JAK mutation) noted but no mechanisms provided. (e.g. FLT3 vs. FLT3-ITD) Would be more on topic then describing classes and age related mortalities

  • Figure 1 would benefit from a list of receptors that associate with the JAKs
  • Later on, the article names specific functions associated with specific STATs, which could also be integrated in figure one to increase the informativity of this figure.

  • Literature lists multiple examples of JAK2/STAT5 activation, including the oncogenic JAK2 V617F variant. The paper notes STAT5 to be mainly driven by JAK1/2. Please clarify.

  • The section of JAK-STAT inactivation falls relatively short in comparison to other sections.

  • The section of JAK-STAT in AML needs more structure. It starts with MPNs (which are not AML) and omits naming MPL and CALR, which eventually all activate STATs.
  • One may just describe the general role of JAK-STAT in hematopoiesis, which inevitably explains how and why aberations in the pathway are so frequent in hematologic malignancies (MPN,MDS, AML). Ways approaching inhibition of JAK-STAT signaling are not named at all in this section, which is odd with the last blurry sentence that they are well recognized targets. The authors could just refer to the next section here instead.

  • The contribution of figure 2 to the review remains unclear to this reviewer.

  • The section regarding JAK inhibitors lacks some outlook and while a serious number of inhibitors has been listed a proper comparison would have been beneficial for readers aiming to choose one for their own work. The same applies for STAT3 and STAT5 inihibitors.
  • Yet, overall this section is appreciated by this reviewer and close to completion.

Author Response

The title should be reconsidered. One main area of JAK-STAT abnormalities are MPNs, which is also reflected in the manuscript.

We apologize for not having chosen a clear title fitting to the manuscript content. As the focus of this review is not on JAK-STAT abnormalities in MPN but rather therapeutic targeting of JAK-STAT signaling in AML we adapted the chapter “Dysregulation of JAK-STAT Signaling in the Pathogenesis of AML” to put less emphasis on MPN and only refer to it in the context of secondary AML. We hope that the reviewer agrees that after these changes the title reflects the content of the review.

Additional proof-reading by a native speaker recommended. Sentence structure sometimes quiet complicated.

We thank the reviewer for this advice. We have proofread the manuscript again, corrected mistakes and edited complicated sentences

Introduction needs more structure. The overall intro of AML is relatively long and would benefit from some shortening. Two hit model (Type I/II class) is outdated, even three classes may not do the job. Too much detail for a JAK/STAT focused review with too little connection to the topic

We apologize for not giving our introduction enough structure. We now have shortened the introduction as suggested by the reviewer and omitted the classification into classes. Instead, examples of frequently mutated genes in AML are presented (line 63).

Most adult AMLs are CN, not fusion; signaling activation is found in all subgroups

We thank the reviewer for this remark. We have now rephrased this section by giving the examples of frequent driver mutations in AML.

Stability of signaling mutations (de novo AML vs. relapse)?

We are aware that this is a very important topic however we find the details of it would be out of the scope of our manuscript.

Constitutive activation of STATs in AML (not by JAK mutation) noted but no mechanisms provided. (e.g. FLT3 vs. FLT3-ITD) Would be more on topic then describing classes and age related mortalities.

We are grateful to the reviewer for raising this concern. We have restructured the chapter about JAK-STAT signaling in AML including more details on the mechanisms how the pathway is activated in a JAK-independent manner (line 212).

Figure 1 would benefit from a list of receptors that associate with the JAKs

We agree with the reviewer that Figure 1 would benefit from additional information on the ligands of JAK-associated receptors. We therefore included a list of common ligands that induce JAK-STAT signaling in the figure legend. We hereby hope that this adaptation meets the reviewer's expectations.    

Later on, the article names specific functions associated with specific STATs, which could also be integrated in figure one to increase the informativity of this figure.

We agree with the reviewer that Figure 1 is a very basic figure providing the principle of JAK-STAT pathway. We feel that inclusion of all functions of specific STATs would contain too much detail to retain the clarity of the figure. We hope that the reviewer agrees with our concern of overfilling the graphic.

Literature lists multiple examples of JAK2/STAT5 activation, including the oncogenic JAK2 V617F variant. The paper notes STAT5 to be mainly driven by JAK1/2. Please clarify.

We stated in the manuscript that STAT5 can be activated by JAK1/3 downstream of common gamma chain-dependent cytokines. We have now added that the activation of STAT5 downstream of e.g. EPO receptor is solely JAK2-dependent (line 178). Thus, we hope that this now better clarifies the relation of the different JAKs and the activation of STAT5.

The section of JAK-STAT inactivation falls relatively short in comparison to other sections.

We agree with the reviewer that it would be desirable to have a longer section of JAK-STAT inactivation.  However, in this review we focus on FDA approvals and clinically relevant inhibitors and because the study of the negative regulation of the JAK-STAT signaling is less intense compared to other aspects of JAK-STAT signaling, this section resulted in a shorter part compared to the others.

The section of JAK-STAT in AML needs more structure. It starts with MPNs (which are not AML) and omits naming MPL and CALR, which eventually all activate STATs. One may just describe the general role of JAK-STAT in hematopoiesis, which inevitably explains how and why aberrations in the pathway are so frequent in hematologic malignancies (MPN, MDS, AML). Ways approaching inhibition of JAK-STAT signaling are not named at all in this section, which is odd with the last blurry sentence that they are well recognized targets. The authors could just refer to the next section here instead.

We have completely restructured this chapter, reflecting the reviewer´s suggestions. We hope that the new version meets the reviewer´s expectations.

The contribution of figure 2 to the review remains unclear to this reviewer.

We have added more description of the figure. We hope that now the appearance of Figure 2 is justified (line 260).

The section regarding JAK inhibitors lacks some outlook and while a serious number of inhibitors has been listed a proper comparison would have been beneficial for readers aiming to choose one for their own work. The same applies for STAT3 and STAT5 inhibitors. Yet, overall this section is appreciated by this reviewer and close to completion.

We thank the reviewer for appreciating this section. We agree with the reviewer that a proper comparison of all listed inhibitors would be very informative. However, the criteria for exact comparison are difficult to define as the inhibitors are tested in different preclinical and clinical settings. We hope that the reviewer understands that sufficient data in comparable cohorts/preclinical settings is not available making a potential comparison rather ambiguous.
